# Antimicrobial Peptides of the Cathelicidin Family: Focus on LL-37 and Its Modifications

**DOI:** 10.3390/ijms26168103

**Published:** 2025-08-21

**Authors:** Olga Evgenevna Voronko, Victoria Alexandrovna Khotina, Dmitry Alexandrovich Kashirskikh, Arthur Anatolievich Lee, Vagif Ali oglu Gasanov

**Affiliations:** Koltzov Institute of Developmental Biology of Russian Academy of Sciences, Moscow 119334, Russia; dim.kashirsckih@gmail.com (D.A.K.); aal1999arth@gmail.com (A.A.L.); gasanovvagif@gmail.com (V.A.o.G.)

**Keywords:** antimicrobial peptides, cathelicidins, LL-37, peptide modification, peptide engineering, LL-37 analog design

## Abstract

Cathelicidins are a family of antimicrobial peptides (AMPs) with broad-spectrum activity and immunomodulatory functions. Among them, the only human cathelicidin LL-37 has garnered significant interest due to its potent antimicrobial, antiviral, antifungal, antiparasitic, and antitumor properties. However, the clinical application of LL-37 is hindered by several limitations, including low proteolytic stability, cytotoxicity, and high production costs. To overcome these challenges, a wide range of design strategies have been employed to modify LL-37 and improve its therapeutic potential. LL-37-based analogs represent promising candidates for the development of next-generation antimicrobial and immunomodulatory therapies. Despite significant progress, further research is required to optimize peptide design, ensure cost-effective production, and validate long-term safety and efficacy. Advances in computational modeling, high-throughput screening, and nanotechnology will play an important role in the translation of modified cathelicidins into clinical practice. This review summarizes key strategies of chemical and structural modifications of LL-37 aimed at enhancing its functional properties. Particular attention is given to truncated and retro-analogs, which preserve or improve biological activity while exhibiting reduced toxicity and increased proteolytic resistance. Furthermore, we highlight the use of nanoscale delivery systems, which facilitate targeted delivery, prolong peptide half-life, and mitigate cytotoxic effects.

## 1. Introduction

The emergence and global spread of multidrug-resistant (MDR) bacteria represent a critical challenge to modern healthcare. It is assumed that antibiotic-resistant infections may cause up to 10 million deaths annually by 2050 [1,2]. As a promising alternative to conventional antibiotics, antimicrobial peptides (AMPs) have garnered significant interest. These peptides are key effectors of the innate immune system, offering protection against a broad range of pathogens, including Gram-positive and Gram-negative bacteria, fungi, and viruses [3].

AMPs serve as the first line of defense against pathogens in the host immune system and exhibit broad-spectrum activity under physiological conditions. According to the CAMPR4 (Collection of Anti-Microbial Peptides) antimicrobial peptide database, as of June 2025, the repository contains 11,827 natural AMP sequences, 12,416 synthetic sequences, and 933 experimentally resolved AMP structures [4]. Despite their sequence and structural diversity, most AMPs share common characteristics: a net positive charge (typically ranging from +2 to +9 due to the high content of positively charged amino acids such as arginine and lysine), a length of fewer than 60 amino acids, a high proportion of hydrophobic residues, and an amphipathic structure with spatially separated hydrophilic and hydrophobic domains [5]. These properties allow AMPs to effectively disrupt negatively charged microbial membranes or to translocate into bacterial cells to interact with intracellular targets such as nucleic acids and chaperones, thus interfering with cell division, RNA synthesis, and gene expression, which ultimately leads to the death of microorganisms [6]. In addition, several AMPs exhibit immunomodulatory or pro-apoptotic activities when interacting with eukaryotic cells [7,8].

One of the most extensively studied classes of AMPs are cathelicidins. These vertebrate-derived cationic peptides are widely expressed across species. In mammals, with species-specific differences in gene number, while primates (including humans), mice, rats, and dogs carry a single cathelicidin-encoding gene, pigs, cows, rabbits, horses, goats, and sheep may harbor up to 11 distinct genes [9,10]. Cathelicidin genes are typically 2 kb in length and consist of four exons: exon 1 encodes the 5′ untranslated region (UTR) and signal peptide; exons 2 and 3 encode the conserved cathelin-like domain; and exon 4 encodes both the mature peptide exhibiting antimicrobial and immunomodulatory activity and the 3′ UTR [11]. While the cathelin domain is evolutionarily conserved across species, the mature *C*-terminal antimicrobial domain is highly variable in both length and amino acid composition [12].

Cathelicidins are primarily produced and secreted by immune and epithelial cells. Neutrophils represent the principal source, where the peptides are stored in secretory and azurophilic granules in an inactive propeptide form [12,13]. This propeptide is structurally linked to the cathelin domain and is separated from its processing enzymes, thereby preventing premature activation and minimizing damage to host cells and tissues [14]. In addition to neutrophils, cathelicidin expression has been observed in skin keratinocytes, epithelial cells of the lungs, gastrointestinal and genitourinary tracts, and mucosal surfaces. For instance, the human cathelicidin hCAP18/LL-37 is found in neutrophil granules and is upregulated during inflammation in keratinocytes and airway epithelium [15].

Activation of cathelicidins occurs via proteolytic cleavage of the propeptide in response to infection or inflammation, resulting in the release of a cathelin-like protein and an antimicrobial peptide [16]. Proteolytic cleavage is mediated by enzymes such as neutrophil elastase, serine proteinase 3, or other enzymes, including prostate-derived gastricsin [17]. This mechanism ensures localized activation of cathelicidins at sites of infection, thereby preserving host tissue integrity and finely modulating the immune response. The mature cathelicidins display considerable variability in length, amino acid sequence, and spatial structure, and exert a broad range of biological functions, including antimicrobial, immunomodulatory, wound healing, and anti-tumorigenic activities [12,15,18].

The only human cathelicidin is LL-37 (AMP, Uniprot P49913; LLGDFFRKSKEKIGKEFKRIVQRIKDFLRNLVPRTES), a 37-residue peptide named after its *N*-terminal leucine-leucine motif. Its precursor, human cationic antimicrobial peptide-18 (hCAP18), is encoded by the *CAMP* gene located on chromosome 3p21.31 (Figure 1A,B). *CAMP* contains four exons and encodes a 170-amino-acid prepropeptide [19,20]. LL-37 is produced by a wide range of cells, including human mesenchymal stem cells, mucosal epithelial cells, neutrophils, monocytes, macrophages, mast cells, NK cells, T and B lymphocytes, adipocytes, and keratinocytes [21,22,23]. Like other cathelicidins, LL-37 is processed through cleavage of the cathelin-like domain to form the active antimicrobial peptide.

LL-37 exerts pleiotropic effects through interactions with various cellular structures, receptors, and signaling pathways. It demonstrates activity against a broad range of pathogens, including Gram-positive and Gram-negative bacteria, viruses, fungi, parasites, and tumor cells, and plays a key role in immune regulation and tissue repair [24,25]. In vivo studies have demonstrated significant antimicrobial efficacy of LL-37 and its analogs in various infection models, underscoring their therapeutic potential. In a systemic *Acinetobacter baumannii* infection model in mice, recombinant LL-37 displayed potent antibacterial activity, with all treated animals surviving and no bacteria in blood [26]. Similarly, in murine models of methicillin-resistant *Staphylococcus aureus* (MRSA) wound infection, LL-37 treatment markedly reduced bacterial loads [27]. Beyond direct bactericidal activity, LL-37 demonstrates immunomodulatory properties in vivo, including improved survival in septic mice via suppression of pro-inflammatory macrophage pyroptosis and stimulation of neutrophil-mediated antimicrobial responses such as the release of neutrophil extracellular traps (NETs) and ectosomes [28].

The involvement of LL-37 has been reported in the pathogenesis of various disorders, including psoriasis, systemic lupus erythematosus, diabetes mellitus, rheumatoid arthritis, cardiovascular pathologies, and multiple types of cancer [29,30,31,32,33]. For instance, it has been shown that leukemic cells (CD20^+^) and myeloid cells (CD68^+^) express hCAP18/LL-37 in patients with chronic lymphocytic leukemia, suggesting that signals from the microenvironment promote its expression [34]. Despite extensive ongoing studies, the full therapeutic potential of LL-37 remains incompletely understood [35].

Nevertheless, clinical application of LL-37 is currently limited by a number of problems, such as cytotoxicity, low stability in biological fluids, rapid degradation by proteases, and high synthesis costs. Therefore, considerable efforts are focused on the development of modified cathelicidins and their structurally optimized analogs with improved characteristics and pharmacological profiles [35,36].

## 2. Structural and Functional Features of LL-37

The human cathelicidin LL-37 is a free amphipathic peptide that contains 37 amino acid residues, including 16 charged residues, resulting in a net positive charge of +6. This cationic nature is critical for interactions with negatively charged bacterial components, such as lipopolysaccharides (LPS) in Gram-negative bacteria and lipoteichoic acids (LTA) in Gram-positive bacteria, thereby facilitating penetration across bacterial membranes [37]. LL-37 can also interact with predominantly polyanionic intracellular targets, including DNA, RNA, ribosomes, and most globular proteins within the bacterial cytoplasm [38]. Thus, alterations in this charge balance, for example, by substitution of basic residues, may result in impaired antimicrobial potency.

Sequence alignment of LL-37 orthologs across vertebrates reveals a conserved distribution of hydrophobic and hydrophilic residues, which underlie the peptide’s amphipathicity and facilitate folding through intramolecular salt bridge formation [39,40]. Studies revealed that LL-37 forms an amphipathic α-helix comprising an *N*-terminal helix containing a di-leucine motif (Leu-Leu) and a *C*-terminal helix extending from residues 2–31, while residues 32–37 remain unstructured, forming a *C*-terminal tail (Figure 1C) [41,42]. In micellar environments, LL-37 adopts an amphipathic “helix-break-helix” conformation, characterized by a helical bend between residues Gly14 and Glu16. This structure is stabilized by a salt bridge between Glu16 and Lys12, as well as a hydrophobic cluster formed by Ile13, Phe17, and Ile20 [43,44]. The *C*-terminal tail is flexible while the helical region is rigid. The hydrophobic surface of LL-37 consists of four individual aromatic Phe side chains (Phe5, Phe6, Phe17, and Phe27) located on the concave side of the peptide and oriented in the same direction. Notably, the hydrophilic Ser9 residue divides the hydrophobic side of LL-37 into two parts [24,43].

Hydrophobic residues, particularly Phe5, Phe6, Phe17, and Phe27 residues and the Ile20–Leu28 segment within the active core region, are highly conserved. In contrast, negatively charged and polar uncharged residues are less strictly conserved [45]. Hydrophobic side chains, including Leu2, Phe5, Phe6, Ile13, Phe17, Ile20, Val21, Ile24, Phe27, Leu28, and Leu31, align along one face of the α-helix, facilitating insertion into the lipid bilayer [43]. Aromatic–aromatic interactions, particularly between Phe residues, form hydrophobic clusters, which may be essential for LL-37 oligomerization [46]. Both Arg23 and all Phe residues have been identified as important determinants of LL-37 molecular interactions, especially with phosphatidylglycerol, a major bacterial membrane lipid [39].

The secondary structure and oligomeric state of LL-37 are influenced by external factors such as pH, salt concentration, the presence of lipids or detergents, and divalent anions, all of which collectively modulate its antimicrobial activity [8,41,47,48]. In solutions, LL-37 exists as an unstructured monomer, but undergoes a structural transition into an α-helical state under conditions that mimic membrane environments, such as elevated peptide concentrations or the presence of salt ions, detergents or lipid-based micelles [49]. LL-37 exhibits remarkable structural plasticity, assembling into antiparallel helical dimers, tetramers, and higher-order oligomers in membrane-mimicking conditions [37,39,47]. Hydrophobic “nests” and aromatic clusters at the dimer interface serve as scaffolds for lipid binding, promoting the formation of stable peptide–lipid complexes and supramolecular fibril-like assemblies that may represent the active membrane-bound state [39]. This conformational adaptability and variability in oligomeric state are thought to underlie the broad range of LL-37 functions [37].

It is important to note that among the classical models describing the mechanisms of bacterial membrane disruption, such as the carpet-like (detergent-type) mechanism, characterized by peptide-induced membrane fragmentation and micelle formation, the toroidal pore model, involving electrostatic interactions between AMPs and membrane lipids, and the barrel-stave model driven by combined hydrophobic and electrostatic forces, LL-37 appears to have a hybrid mode of action. Specifically, it exhibits features of both the toroidal pore, through lipid engagement, and the carpet-like mechanism, via lipid extraction from the bilayer [8,39,49]. The amphipathic α-helices of LL-37 first associate with the membrane surface through electrostatic attraction and then penetrate into the hydrophobic core, stabilized by interactions with lipid acyl chains [44,49,50]. This dual mode not only perturbs the bilayer but also alters its mechanical properties, with the bilayer order influencing the depth of insertion and the extent of disruption. However, the accumulating evidence suggests that the action of LL-37 is more intricate than these classical models imply [39].

The *C*-terminal helix is primarily responsible for antimicrobial and anticancer activity, whereas the *N*-terminal mediates chemotaxis, proteolytic resistance, and contributes to hemolytic activity [46]. The flexible *C*-terminal tail is essential for tetramerization [51]. Mutagenesis and truncation studies of LL-37 have shown that while the initial four *N*-terminal residues are not essential for antimicrobial activity and involved in peptide oligomerization, the first twelve are critical for chemotaxis [41]. The *N*-terminal di-leucine motif also initiates autophagy in macrophages, and its substitution with di-alanine or removal from the peptide sequence abrogates this function without compromising antimicrobial efficacy [52].

Membrane permeabilization assays have demonstrated that LL-37 and its truncated analog LL-32 exhibit concentration-dependent disruption of bacterial membranes [53]. LL-32 shows greater activity at high concentrations, while both peptides exhibit similar activity at lower concentrations. These effects are attributed to the VPRTES motif within the *C*-terminal tail [53]. According to computational modeling, this motif may facilitate LL-37 insertion into the membranes and plays an important role in the control of the peptide oligomerization on the membrane surface.

## 3. Endogenous Post-Translational Modifications of LL-37

Prior to discussing synthetic modification strategies, it is essential to consider the natural post-translational modifications (PTMs) of LL-37 and their influence on peptide function. LL-37 undergoes several PTMs in vivo, including citrullination, carbamylation, acetylation, and formylation [52,54,55,56]. These modifications alter its structure and function and have been implicated in the pathogenesis of various inflammatory and autoimmune diseases, including systemic lupus erythematosus (SLE) [57]. PTMs may alter the properties of LL-37, turning it into a potential autoantigen and enhancing the immune response. Notably, PTMs of LL-37 in vivo can be caused by neutrophils and macrophages, which in turn are its significant sources [52].

Citrullination is catalyzed by peptidylarginine deiminases (PADs), which convert Arg residues into citrulline [58]. Carbamylation is a non-enzymatic PTM that results in the conversion of Lys and Leu residues into homocitrulline [59,60]. This process depends on the reaction of cyanate with primary amines, and is often stimulated by myeloperoxidase (MPO) in activated neutrophils. Both PTMs are closely associated with the neutrophilic inflammation observed during SLE and can occur simultaneously, leading to the formation of both citrullinated LL-37 (cit-LL37) and carbamylated LL-37 (carb-LL37) [55,57]. Functionally, citrullination disrupts the α-helical structure of LL-37, which is accompanied by a complete loss of antimicrobial activity against *Escherichia coli*. In contrast, *N*-terminal acetylation and formylation preserve the helical structure and antimicrobial function [52]. This phenomenon may indicate that some of the *N*-terminal modifications that do not disrupt the full structure of the peptide do not affect its ability to kill bacteria.

Citrullination and carbamylation modulate immunological roles of the LL-37. Natural LL-37 is known to act as T-cell/B-cell autoantigen in psoriatic disease and SLE [57]. Cit-LL37 enhances antigen presentation while suppressing type I interferon (IFN-I) production and B-cell maturation [55,57]. In contrast, carb-LL37 retains the ability to activate innate immune cells such as plasmacytoid dendritic cells (pDCs) and B cells, and promotes autoantibody production. These modifications not only alter the functional properties of the LL-37, but also make it an important element in the pathogenesis of inflammatory diseases [55].

The cellular environment is also important for LL-37 functions. Natural PTMs may influence the ability of LL-37 to induce autophagy in macrophages, a process critical for infection control and resolution of inflammation [52,56,61]. Native LL-37 efficiently induces autophagy, whereas its formylated and acetylated variants lose this ability. For example, neutrophil-derived LL-37 is unable to induce autophagy due to *N*-terminal acetylation and formylation [52]. Moreover, mass spectrometry analysis has revealed that macrophage-derived LL-37 remains largely unmodified, whereas neutrophil-derived LL-37 undergoes extensive *N*-terminal modifications [56].

Understanding natural PTMs and their functional implications (Table 1) is critical for the development of therapeutic strategies aimed at enhancing or mimicking LL-37 function while avoiding immunopathological consequences.

## 4. Strategies for LL-37 Functional Optimization

Although cathelicidins, particularly LL-37, represent promising templates for the development of novel antimicrobial agents, their clinical application is hindered by several limitations, including proteolytic instability, insufficient selectivity towards host cells, high production costs associated with their full-length sequence, and significant cytotoxicity.

LL-37 can damage the plasma membranes of erythrocytes, lymphocytes, and fibroblasts in vitro at concentrations similar to those required for antimicrobial activity [40]. Elevated LL-37 levels (>100 μM) have been reported in patients with ulcerative colitis, psoriasis, rosacea, and chronic periodontitis, potentially contributing to host cell damage and apoptosis at sites of inflammation [35]. Several strategies have been proposed to mitigate these effects while preserving antimicrobial potency.

Maintaining the stability of LL-37 in biological environments remains a significant issue, as cathelicidins are susceptible to proteolytic cleavage and unstable under high salt concentrations [62]. The main approaches to enhance its stability include: (1) incorporation of helix-stabilizing residues; (2) cyclization via disulfide bridges; (3) substitution of L-amino acids for D-enantiomers to provide resistance to proteolytic degradation.

Given that the antimicrobial activity of cathelicidins is largely determined by amino acid sequence, charge distribution, hydrophobicity, and amphipathicity [62], current modification strategies aim to enhance antimicrobial and other biological efficacy, reduce cytotoxic and hemolytic activity, and increase proteolytic stability of LL-37 and its derivatives both in vitro and in vivo. These approaches include generating truncated forms or retro-analogs, amino acid sequence modifications, modulation of physicochemical properties (such as hydrophobicity and net charge), and encapsulating peptides within nanostructured delivery systems. The following sections detail key strategies for LL-37 modification.

### 4.1. Shortened Derivatives: Truncated LL-37 Analogs

One promising approach to mitigate these issues involves the truncation of the LL-37 sequence, particularly by removing hydrophobic *N*-terminal residues responsible for hemolytic activity [63]. Truncation of LL-37 reduces toxicity, limits protease cleavage sites, and improves selectivity and stability. In human sweat, LL-37 is processed by kallikreins 5 and 7 into truncated peptides such as RK-31 (residues 7–37), KS-30 (residues 8–37), and KR-20 (residues 18–37), all of which exhibit antimicrobial activity [64]. In addition to these truncated peptides of natural origin, there are a number of analogs of LL-37, which are active fragments of the full peptide, such as FK-13 (residues 17–29), FK-16 (residues 17–32), as well as the mentioned above KR-12 (residues 18–29), and others (Figure 2A).

KR-12 (KRIVKLIKKWLR) is one of several minimal LL-37 fragments that retain antimicrobial activity while demonstrating significantly reduced cytotoxicity and hemolytic activity [65]. With a net charge of +5 and amidated *C*-terminus, KR-12 exhibits improved resistance to carboxypeptidases. It is effective against *E. coli* but lacks activity against *S. aureus* [43]. In turn, FK-13, another minimal LL-37 fragment with one additional amino acid compared to KR-12, exhibits activity against human immunodeficiency virus (HIV) [62]. Notably, KR-12 can also induce apoptosis in human breast cancer cells [66]. It has been suggested that the amphipathicity of KR-12 (visualized as helical wheel projection) demonstrates superior amphipathic helicity compared to LL-37, which may thus contribute to its enhanced antimicrobial activity [67]. Heliograms of LL-37 and its selected analogs are presented in Figure 2B. This type of projection allows identification of hydrophobic and hydrophilic faces of an α-helical peptide and planning of target modifications to stabilize the structure [68,69].

Other minimal fragments, such as FK-13 and FK-12 (which lacks the terminal Arg residue found in FK-13), inhibit the growth and biofilm formation of *Staphylococcus epidermidis* [70]. In turn, FK-16 (FKRIVQRIKDFLRNLV) induces caspase-independent apoptotic cell death via the p53–Bcl-2/Bax cascade and autophagy in cancer cells, highlighting its anticancer potential [71,72].

GF-17 (GFKRIVQRIKDFLRNLV) and GI-20 (GIKEFKRIVQRIKDFLRNLV) contain the core antimicrobial domain, maintain a highly helical structure, and are often used as templates for further modifications. Their derivatives like 17BIPHE2, GF-17d1-3, and GI-20d, containing unnatural aromatic residues (e.g., Phe17 and Phe27 substituted by biphenylalanine) or composed of D-amino acids, respectively, exhibit broad-spectrum antimicrobial and antiviral activities in vivo, including inhibition of Ebola virus by blocking cathepsin B [73]. Substitution of Gly to Trp in 17BIPHE2 enhances binding to bacterial membranes and DNA, increasing activity against MRSA, while the Arg to Glu substitution retains its activity against MRSA but reduces cytotoxicity [51].

OP-145 (Ac-IGKEFKRIVERIKRFLRELVRPLR-NH_2_), previously known as P60.4Ac, is an acetylated and amidated synthetic peptide derived from LL-37 with LPS and LTA binding capacity. OP-145 has been developed as an ear drop for the treatment of chronic bacterial middle ear infections, and has demonstrated potent effects against MRSA in epithelial models [74,75,76]. In a bronchial epithelial model, it reduced MRSA viability by 90%, and in epidermal models, it eliminated 99% of planktonic bacteria and 85% of biofilm-associated bacteria without detectable cytotoxicity toward epithelial cells and keratinocytes [75]. Although OP-145 exhibits improved antimicrobial activity compared to LL-37, its efficacy diminishes in complex biological fluids [77]. Nevertheless, it has shown robust in vivo activity, significantly reducing *S. aureus* load when administered using implants in infected murine models [77]. Further modification, including substitution of a hydrophobic amino acid with a cationic residue in the hydrophobic face of OP-145, enhanced membrane selectivity, while reducing antimicrobial activity against *S. aureus* and selected Gram-negative strains in vitro, yet without toxicity to human dermal fibroblasts [76].

Another example is SAAP-148 (Ac-LKRVWKRVFKLLKRYWRQLKKPVR-NH_2_) derived from a synthetic antibacterial and antibiofilm peptide (SAAP) library [78,79]. SAAP-148, like OP-145, is a 24-mer antimicrobial peptide that was designed from consensus sequences of the LL-37 homolog and is active against both Gram-positive and Gram-negative bacteria [80]. SAAP-148, applied as a part of a supramolecular coating for titanium implants, retains antimicrobial activity and effectively kills antibiotic-resistant *S. aureus*, *A. baumannii*, and *E. coli*, while demonstrating minimal toxicity to human dermal fibroblasts [81]. Nevertheless, the ability of SAAP-148 and OP-145 to permeabilize *E. coli* membranes does not fully correlate with their bactericidal activity. Their mechanism of action involves interaction with the bacterial surface, disruption of the cytoplasmic membrane, perturbation of lipid packing, and membrane depolarization via an LPS-independent mechanism [80]. Furthermore, highly cationic SAAP-148 kills *Enterococcus hirae* at significantly lower concentrations than OP-145 and causes stronger membrane depolarization and permeabilization effects at both cellular and molecular levels [82].

SAAP-148 and its variant SLP-51 (Ac-KKRVKKRSFKLSKSYWRSLKKPVR-NH_2_), containing two hydrocarbon staples and the *C*-terminal Arg glycosylation, show enhanced antibiofilm and antimicrobial efficacy along with improved proteolytic resistance [83]. Moreover, SLP-51 demonstrates reduced hemolytic activity compared to the parent peptide. In both skin wound and drug-resistant pneumonia mouse models, SLP-51 exhibits potent therapeutic effects against MRSA and *K. pneumoniae* infections in vivo, along with marked attenuation of inflammatory damage.

It is also worth mentioning that a number of LL-37-derived analogs possess antifungal and antiprotozoal properties. Fragments such as LL37-1 (GRKSAKKIGKRAKRIVQRIKDFLR) amidated at the *C*-terminal position, ACL-37-1 (AC-1; RKSKEKIGKEFKRIVQRIKDFLR) with an acetylated *N*-terminal domain, ACL-37-2 (AC-2; GRKSAKKIGKRAKRIVQRIKDFLR), which is acetylated and amidated at the C-terminus, and DL-37-2 (D (d-PHE); GRKSAKKIGKRAKRIVQRIKDF_d_LR) containing Phe substituted with the D-enantiomer, have demonstrated significant activity against *Candida* spp. [84], while KR-12, KR-20, and KS-30 have shown efficacy in compromising the viability and membrane integrity of *Entamoeba histolytica* trophozoites [85]. Additional lipidation strategies have enhanced the antifungal activity of KR-12 against multiple *Candida* species, including *C. albicans*, *C. glabrata*, *C. tropicalis*, and *C. lipolytica* [86]. An important feature of these peptides is their structural optimization for increased protease resistance and reduced hemolytic potential compared to native LL-37.

Despite their advantages, truncated peptides often exhibit reduced stability. Thus, current solutions include head-to-tail cyclization and incorporation of D-amino acids, which restore or enhance biological activity and proteolytic resistance [73,87,88,89].

### 4.2. Sequence-Reversed Variants: Retro-Analogs of LL-37

Peptides with reversed amino acid sequences, known as retro-analogs, represent a promising strategy for LL-37 modification [90]. Sequence inversion preserves amphipathicity and net charge while enhancing resistance to proteolytic degradation. Moreover, increased hydrophobicity in retro-analogs has been associated with enhanced antimicrobial activity [90].

SE-33 peptide (SETRPVLNRLFDKIRQVIRKFEKGIKEKSKRFF) is a retro-analog of LL-37 fragment (residues 5–37) that retains the amphipathic α-helix and exhibits potent activity against bacteria such as *E. coli* (including drug-resistant strains), *S. aureus*, *Lactobacillus casei*, and fungi (e.g., *C. albicans*) [91,92,93]. The minimum inhibitory concentration of SE-33 for fungi and yeast ranged from 31.2 to 1024 μg/mL, while for *S. aureus* it was 12.5 μg/mL, with the minimum bactericidal concentration from 20 μg/mL for *E. coli*.

Modulation of hydrophobicity in retro-analogs has been shown to significantly influence biological activity. It has been hypothesized that the *N*- and *C*-terminal amino acid residues play a significant role in the difference between the hydrophobicity of peptides and their retro-analogs [90]. For KR-12, two retro-analogs, including retro-KR12-NH_2_ (RLFDKIRQVIRK-NH_2_) and its acetylated form Ac-retro-KR12-NH_2_ (Ac-RLFDKIRQVIRK-NH_2_) were synthesized [94]. Analysis revealed differences in calculated hydrophobicity coefficients for Lys and Arg in these retro-analogs compared with the parent KR-12, indicating their increased hydrophilicity, as confirmed by high-performance liquid chromatography retention profiles [94].

In order to increase selectivity toward bacterial membranes and improve antibacterial and hemolytic activities, additional lipidation at the *N*-terminus with octanoic acid was applied to the retro-KR12-NH_2_, which resulted in the generation of C_8_^α^-retro-KR12-NH_2_ (C_8_-RLFDKIRQVIRK-NH_2_) and Retro-KR12-C_8_^ε^-NH_2_ (RLFDKIRQVIRK(C_8_)-NH_2_) [86,94]. These lipidated retro-peptides carried a net charge of +4 and displayed reduced hemolysis, though their overall antimicrobial potency was lower than that of the parent molecules [86,94]. Notably, C_8_^α^-retro-KR12-NH_2_ achieved a markedly higher selectivity index (>21) than Retro-KR12-C_8_^ε^-NH_2_ (1.52) and a slightly higher one than parental peptides. Additionally, C_8_^α^-retro-KR12-NH_2_ exhibited superior antibacterial properties against *S. aureus* strains, while showing more moderate activity against pathogens from the ESKAPE group, indicating pronounced efficacy against Gram-positive bacteria.

Taken together, these findings underscore the therapeutic potential of retro-analogs as candidates for next-generation antimicrobial peptide design.

### 4.3. Key Strategies for Modulation of LL-37 and Its Derivatives

#### 4.3.1. Rational Sequence Engineering: Site-Specific Amino Acid Substitutions

As previously discussed, distinct regions within the LL-37 amino acid sequence are responsible for specific biological functions, including membrane interaction, bacterial cell penetration, and binding to cytosolic components, thereby mediating antimicrobial activity. Alterations within these regions, whether through residue substitution, incorporation of new residues, or truncation of the original sequence, can substantially influence peptide function. Shorter LL-37 derivatives have demonstrated improved penetration of bacterial membranes and prevention of biofilm formation [67].

Selective substitution of hydrophobic residues has been shown to reduce cytotoxicity without significantly impairing the antimicrobial activity of the peptide [95]. For example, positional Q and K mutants of LL-37, in which hydrophobic residues were replaced with mildly hydrophilic amino acids such as Glu and Lys, respectively, exhibit lower hemolytic activity while maintaining the ability to penetrate human breast cancer cells [96]. Due to its small size, LL-37 truncated analogs are highly susceptible to modifications. For instance, substitution at key positions, such as Gln22 and Asp26 with Lys, leads to an 8-fold increase in KR-12 antimicrobial activity [97].

Furthermore, targeted amino acid substitutions in LL-37 and its analogs have been explored to enhance their antitumor activity. For instance, the synthetic peptide FF/CAP18 (FRKSKEKIGKFFKRIVQRIFDFLRNLV), in which Glu and Lys are substituted by Phe, has been shown to inhibit proliferation and promote apoptosis in colon cancer cells [98,99]. Substitution of Ser9 with Ala or Val in native LL-37, followed by truncation to generate LL-23, results in increased antibacterial and immunosuppressive activity [100]. In addition, substitution of positively charged amino acids at 2, 6, 12, 17, 19, and 23 positions with His residues in LL-37 increases its selectivity towards binding a tumor membrane [101].

Various structural modifications can be applied to significantly enhance the proteolytic stability of LL-37 and its derivatives. Such modifications may include *N*-terminal Gly insertion to prevent aminopeptidase cleavage, and *C*-terminal amidation to reduce carboxypeptidase sensitivity [69]. Another common strategy involves the substitution of L-amino acids with their D-enantiomers, particularly at protease-sensitive positions. For instance, the LL-37 analog GF-17d3, in which residues 20, 24, and 28 are replaced by D-amino acids, exhibits a 10-fold increase in serum stability while maintaining antibacterial efficacy following incubation with chymotrypsin [102]. Substitution of L-amino acids for D-forms in GF-17d (the full D-enantiomer of the LL-37 fragment) makes it completely resistant to serum proteases [103]. In turn, L-Lys to D-Lys substitution increased resistance of the FEG peptide to NaCl and MgCl_2_ [104]. These modifications are crucial for peptide resistance to proteases and maintaining activity under rigid conditions.

Incorporation of non-canonical amino acids (e.g., aromatic or β-amino acids) can enhance antimicrobial activity of AMPs through cation-π interactions, which facilitate peptide incorporation into lipid membranes. The massive side chain of aromatic amino acids also contributes to greater loosening of the lipid layer [105]. For example, arylation of tryptophan residues improves antimicrobial activity without increasing the hemolytic effect [106]. Replacement of tryptophan with β-naphthylalanine increases the resistance of AMPs to high salt concentrations and promotes their translocation across the membranes of Gram-negative bacteria [107]. In turn, incorporation of β-amino acids in the LL-37 sequence further contributes to resistance against enzymatic degradation [108].

It should be noted that some substitutions may have a negative effect on LL-37 function. For example, proline substitutions destabilize the α-helical structure of the peptide, reducing its interaction with phospholipid membranes, and have been associated with decreased toxicity [109]. However, there are reports indicating that substitution of Ile20 with Pro in LL-34, a truncated analog of LL-37, does not affect its ability to penetrate bacterial or mammalian membranes and does not alter its cytotoxicity or antimicrobial activity. Nonetheless, this modification abolished cytokine-inducing activity, thereby altering its immunomodulatory function [110].

#### 4.3.2. Strategies for Modulating Hydrophobicity, Amphipathicity, and Net Charge

A balanced combination of cationicity, hydrophobicity, and amphipathicity is considered critical for the membrane-disruptive activity of most AMPs, including LL-37 [111]. These properties promote electrostatic interactions of AMPs with negatively charged microbial membranes, followed by penetration into the hydrophobic lipid bilayer [67,112]. Amphipathicity, defined as the spatial separation of hydrophilic and hydrophobic domains, correlates with both antimicrobial potency and cytotoxicity [113]. Notably, an increase in amphipathicity tends to elevate hemolytic activity more than bactericidal activity [56]. Hydrophobic residues, which typically constitute 40–60% of the sequence, enable interaction with the lipid bilayer of membranes. In LL-37, reducing hydrophobicity has been shown to decrease cytotoxicity [43,46]. However, an excessive increase in hydrophobicity leads to a decrease in specificity and an enhanced toxicity of AMPs towards eukaryotic cells. Moreover, a positive net charge (+3 to +6) facilitates electrostatic interaction with negatively charged bacterial membranes.

Structural studies indicate that truncated LL-37 derivatives such as KE-18 and KR-12 exhibit stronger antimicrobial activity against bacteria and fungi (*C. albicans*, *S. aureus*, and *E. coli*) and improved LPS- and LTA-binding capacity compared to the full-length parental peptide [67]. The observed decrease in cationicity of KE-18 and KR-12 compared to LL-37 may have been compensated by a slightly higher hydrophobic ratio. These truncated peptides also displayed superior amphipathic helical structures that may facilitate membrane interaction. Lower hydrophobicity has been suggested to associate with reduced cytotoxicity and decreased selectivity towards mammalian cells, as indicated in multiple studies [114,115,116,117,118].

In order to investigate the effect of peptide hydrophobicity on cytotoxicity towards eukaryotic cells, several novel derivatives of KR-12 (KR-12-a1 to KR-12-a6), with optimized structures, were generated [115]. KR-12-a1 was designed through a substitution of Phe10 to Trp, while KR-12-a2 and KR-12-a3 were generated by substituting Asp9 or Gln5 with Lys to increase the net positive charge. KR-12-a4, KR-12-a5, and KR-12-a6 were derived from KR-12-a3 by introducing additional Leu residues to enhance the hydrophobic angle. Among these, KR-12-a5 and KR-12-a6, the most hydrophobic analogs, exhibited superior inhibition of LPS-induced TNF-α production and demonstrated enhanced LPS-binding capacity, indicating significant anti-inflammatory potential. Furthermore, KR-12-a2, KR-12-a3, and KR-12-a4 displayed significantly improved selectivity for bacterial cells over erythrocytes while maintaining anti-endotoxic activity, thereby exhibiting reduced cytotoxicity toward mammalian cells compared to native LL-37. Moreover, all the mentioned KR-12 analogs showed potent antimicrobial activity against MRSA.

A decrease in hydrophobicity is generally correlated with lower cytotoxicity and improved selectivity of LL-37 derivatives [116,117]. For instance, incorporation of D-amino acid into KR-12-a5 enhanced cell selectivity by 2.6–13.6-fold compared with the parental peptide. In particular, KR-12-a5(6-^D^L), containing a D-Leu6 substitution at the polar–nonpolar interface, exhibited the lowest hydrophobicity, minimal hemolytic activity, and preserved antimicrobial potency among all tested analogs [116]. Similarly, a series of Arg-, Trp- or Arg/Trp-substituted variants of the FK-13 demonstrated increased positive net charge and hydrophobicity, improved cell selectivity with no detectable hemolytic activity, retained anti-inflammatory activity accompanied by high LPS-neutralizing activity, and 2.6- to 5.8-fold higher antimicrobial activity relative to unmodified FK-13 [117]. In turn, truncated LL-37 fragment GF-17 forms a symmetric amphipathic structure with relatively low hydrophobicity, enabling effective pathogen and cancer cell targeting while minimizing cytotoxicity toward human cells [118]. This study also emphasizes the role of the cationic residue Asn30 in the early adsorption, and Phe17, Phe27, Leu28, Leu31, and Val32 hydrophobic residues in the membrane insertion of the LL-37 analog, regardless of initial orientation.

*C*-terminal amidation and substitution of neutral or negatively charged residues with positively charged amino acids (Lys or Arg) can further increase net charge and enhance the affinity of peptides for the membrane [69]. However, excessive cationicity may increase cytotoxicity and disrupt amphipathic balance, thereby reducing peptide specificity. Modulation of charge distribution, particularly in the *N*-terminal region (e.g., in the KE-18), enhances binding to bacterial LPS [67].

Additionally, several studies have indicated that cathelicidin derivatives with increased cationicity demonstrate enhanced antibacterial properties, as stronger electrostatic interactions result in higher affinity for bacterial membranes [119,120]. Conversely, reducing the overall charge (from +6 in native LL-37 to +4 or +3 in some of its derivatives) also reduces toxicity [86,94]. Moreover, peptides with high cationicity and hydrophobicity exhibit greater affinity for the zwitterionic membranes of human red blood cells, resulting in increased hemolytic activity [121]. Thus, a careful balance between the charge and hydrophobicity of the peptide must be maintained.

#### 4.3.3. Conformational and Structural Modifications: Cyclization, Hybridization and Lipidation

Cyclization of cathelicidins via disulfide bridges or β-turn formation has been shown to enhance proteolytic stability in vivo. For instance, the cyclic analog of PST13-RK became 2-times more stable in blood plasma, while maintaining activity against *E. coli* [122]. In the AdCath, a cathelicidin from the skin of Chinese giant salamanders, *Andrias davidianus*, disulfide bridges stabilize the β-sheet structure [123]. For LL-37 and its truncated analogs, cyclization through disulfide bridges or dimerization combined with backbone cyclization (e.g., CD4-PP, derived from KR-12) improved conformational rigidity and enhanced proteolytic resistance under physiological conditions [87,88,89].

Peptide hybridization and dimerization can also have a positive effect on its antimicrobial activity. A LL-37/magainin II fusion displayed a 3-fold reduction in minimum inhibitory concentration against *Porphyromonas gingivalis* compared to the parental sequences [124]. In turn, LL-37 conjugation with immunostimulatory CpG oligodeoxynucleotides has been reported to enhance activation of innate immunity, thus leading to the control of ovarian tumors [125].

Enhanced antimicrobial activity against Gram-negative bacteria can be achieved through lipidation strategies, such as the conjugation of acyl chains to the peptide structure in order to enhance interaction with lipid membranes [126]. Additionally, conjugation with fatty acids such as lauric or myristic acid improves stabilization of the helical structure, as well as thermal stability of peptides [127]. Such lipidation enhances bacterial membrane interactions by increasing the hydrophobicity of AMPs. Myristoylated derivatives of KR-12 (Myr-KR-12N and Myr-KR-12C) exhibit increased antimicrobial activity and augmented membrane interaction, and form self-assembling nanoparticles [128]. Myr-KR-12N, in particular, has shown significant LPS-binding capacity, effective reduction in inflammation in vitro, and the ability to protect mice from *E. coli*-induced sepsis compared to the conventional antibiotics. Conjugation of KR-12 peptide and its analogs with additional octanoic, 2-buthyloctanoic, or 4-phenylbenzoic acid residues demonstrated improved activity against *E. coli*, *S. aureus*, *Klebsiella pneumoniae*, *Streptococcus pneumoniae*, *Acinetobacter baumannii*, and *Pseudomonas aeruginosa* due to increased membrane affinity [94].

Another advanced approach involves all-hydrocarbon stapling, wherein covalent alkyl bridges are introduced between side chains of α,α-disubstituted amino acids (e.g., S-pentenylalanine) using the ring-closing metathesis method [129]. This modification stabilizes the α-helical structure, thereby preventing the helix from unfolding, increasing protease resistance and structural integrity. To enhance KR-12 functionality, a series of derivatives with one- or two-helical wheels stapled at either position of *i*, *i* + 4 or *i*, *i* + 7 were synthesized [130]. Among them, KR-12(Q5, D9), stapled at the Glu and Asp residues, exhibited both increased positive net charge and enhanced α-helicity. These structural reinforcements not only improved antibacterial and anti-inflammatory efficacy but also conferred resistance to proteolytic degradation. In vitro, the peptide displayed superior mammalian cell compatibility and a significantly higher therapeutic index. In a murine wound model infected with *E. coli*, KR-12(Q5, D9) effectively eliminated bacteria while accelerating wound closure and tissue regeneration [130].

While hydrophobic staples stabilize α-helical conformation, they can also induce non-selective membrane disruption and elevate cytotoxicity towards mammalian cells, resulting in a low therapeutic index [131,132]. A promising alternative involves incorporating hydrophilic modifications in the stapled peptides, such as *N*- or *C*-terminal glycosylation, which can improve peptide activity while mitigating hemolysis and toxicity [133]. This strategy was applied to SAAP-148 (SLP-0) [83]. Using a peptide library of SAAP-148 derivatives, an optimized analog, SLP-51, was identified and generated through all-hydrocarbon stapling, Lys canning, double-stapling, and Arg *N*-glycosylation. Compared with the linear SLP-0, glycosylated double-stapled SLP-51 demonstrated up to 35-fold enhanced antimicrobial potency against multiple pathogens (*S. pneumoniae*, *S. aureus*, *P. aeruginosa*, *K. pneumoniae*), greater stability in human plasma, and minimal hemolysis. Molecular dynamics simulations revealed that SLP-51 penetrates bacterial membranes significantly earlier (≈300 ns) than its linear counterpart, underscoring the impact of double-stapling on stability and membrane translocation. Moreover, in both skin wound and drug-resistant bacterial pneumonia models, SLP-51 showed a potent therapeutic effect in treating both MRSA and *K. pneumoniae* infection in mice and significant improvement of inflammatory injury.

The key LL-37 modification strategies discussed in Section 4.1, Section 4.2 and Section 4.3 are summarized in Table 2, highlighting the structural principles, representative examples, and resulting functional benefits. Despite current limitations, cathelicidins, particularly human LL-37 and its derivatives, remain an important focus in antimicrobial drug development. Rational modification strategies continue to expand their therapeutic potential.

## 5. Nanoscale Delivery Platforms for LL-37 and Its Derivatives

In addition to structural modification, an emerging strategy to improve the therapeutic potential of LL-37 is the use of nanoscale delivery systems (Figure 3) [134]. Encapsulation of peptides into liposomes, polymeric nanoparticles, or hydrogels enhances their stability and enables controlled release. These nanocarriers typically possess a net positive surface charge, facilitating selective electrostatic interaction with negatively charged bacterial membranes. Moreover, lipid-based encapsulation of peptides provides protection against proteolytic degradation.

To be effective carriers, nanoparticles must meet several criteria: they should be biocompatible, non-toxic, structurally stable, and capable of targeted delivery to specific tissues or cells [135]. Various nanomaterials have been investigated for peptide delivery, including systems designed for anticancer applications [136,137]. For instance, LL-37 loaded onto zinc oxide nanoparticles (ZnO NPs) significantly inhibited the proliferation of the BEAS-2B human lung cancer cell line [138]. Similarly, loading LL-37 onto magnetic nanoparticles enhanced its anticancer efficacy, suggesting that nanoparticle-mediated delivery may facilitate the use of LL-37 in oncological studies [139,140].

Peptide immobilization on biomaterials is known to extend its bioactivity and prolong its half-life. For instance, polymeric carriers such as poly(lactic-co-glycolic) acid (PLGA) nanoparticles offer sustained and controlled release, increasing the half-life of LL-37 from 2 to 24 h [141]. PLGA–LL-37 nanoparticles also promote keratinocyte migration in vitro, retain antimicrobial activity against *E. coli*, and accelerate wound healing in mice. Additionally, incorporation into mesoporous silica nanoparticles preserves LL-37 activity for over 50 days [142], while calcium phosphate (CaP) nanoparticles protect it from enzymatic degradation [143].

Another way to overcome the disadvantages of limited bioavailability, high cytotoxicity, and low stability of cathelicidins is the use of nanogels. Nanogels and hydrogels, particularly those containing hyaluronic acid, are promising for local application at wound sites, supporting tissue repair and enabling environment-responsive release. For instance, nanogels containing modified hyaluronic acid with a long-chain lipid (C18) have demonstrated enhanced delivery, reduced toxicity, and increased membrane permeability of a snake-derived cathelicidin Ab-Cath [144]. In vitro analyses showed that encapsulation of the Ab-Cath retained its antimicrobial properties, reduced hemolytic and cytotoxic activity, while the selectivity level increased by 16.8 times. Another member of the cathelicidin family, the peptide Hc-CATH, derived from the venom of the sea snake *Hydrophis cyanocyntus*, covalently bound to the titanium implant surfaces, has been shown to prevent biofilm formation [145]. A similar approach was applied to the LL-37. A hybrid hydrogel loaded with LL-37 and hyaluronic acid accelerated the healing of diabetic ulcers, while thermoresponsive LL-37-loaded hydrogels demonstrated enhanced antiangiogenic and antitumor effects [146,147]. Furthermore, encapsulation of SAAP-148 in nanogels composed of octenyl succinic anhydride-modified or oleylated hyaluronic acid significantly reduced cytotoxicity towards eukaryotic cells in vivo and enhanced antimicrobial activity [148,149]. In another study, cryogels incorporating KR-12 and hyaluronic acid released the peptide in response to bacterial hyaluronidase, enabling the development of dressings for wounds complicated by antibiotic-resistant infections [150].

Lipid-based nanocarriers (liposomes) also represent a widely investigated nano-delivery platform. Both hydrophilic and hydrophobic drugs can be efficiently delivered to tumor sites using fusogenic liposomes that avoid endosomal degradation [151,152]. Encapsulation of LL-37 in lipid vesicles enhances its therapeutic potential by extending its duration of action and reducing cytotoxicity. In murine skin infection models, LL-37 encapsulated in phosphatidylcholine liposomes increased antimicrobial activity and was highly effective against *S. aureus* [153]. A chitosan-coated liposomal formulation with the LL17-32 (LL-37 fragment) exhibited only weak antimicrobial activity against *P. gingivalis*, likely due to limited peptide release [154]. Nonetheless, this delivery approach may serve as a platform for sustained-release formulations.

Nanodelivery systems also enable synergistic effects when combining LL-37 with other antimicrobial or host-directed peptides. For example, solid lipid nanoparticles co-loaded with LL-37 and serpin A1 significantly accelerated wound healing [155]. Additionally, LL-37 can be combined with synthetic analogs such as ceragenins (e.g., CSA-13, CSA-131). Their co-delivery with LL-37 via gold-coated core–shell magnetic nanoparticles reduced the minimal inhibitory concentration of LL-37 by 64- and 32-fold against *S. aureus* and *P. aeruginosa*, respectively [156].

Further strategies for LL-37 and its derivatives delivery include peptide–nanofiber systems. A modified truncated analog, 17BIPHE2, was successfully embedded into the core of pluronic nanofibers (F127/17BIPHE2-PCL), which demonstrated potent activity against MRSA [157]. Similarly, complexes of 17BIPHE2 with nanofibers exhibited bactericidal effects against clinical strains of *K. pneumoniae* and *A. baumannii* in vitro, without cytotoxicity towards skin cells or monocytes. In a biofilm-containing chronic wound model based on type II diabetic mice, 17BIPHE2-containing nanofiber dressings reduced MRSA load by 5-fold. Thus, these nanofiber-based systems show potential for incorporation into biodegradable wound dressings aimed at treating biofilm-associated infections.

## 6. Future Prospects

Despite significant progress in the development of modified AMPs, the translation of LL-37 and its analogs into clinical use remains limited by a number of unresolved technological and methodological issues.

First, large-scale and cost-effective production methods must be developed. The high complexity and expense of traditional chemical peptide synthesis necessitate a shift toward peptide production using recombinant technologies, which offer greater scalability and lower costs [91,92,121,158]. For instance, an orally administered recombinant *Lactococcus lactis* strain engineered to express human LL-37 was recently shown to exhibit antiviral effectiveness in vivo. In animal models, it reduced mortality in piglets infected with porcine reproductive and respiratory syndrome virus and prolonged survival in chickens infected with Newcastle disease virus, supporting the potential of recombinant microbial platforms for therapeutic LL-37 delivery [158]. In addition, truncation of the peptide to its minimal active core not only enhances selectivity but also reduces manufacturing costs. We previously developed a method for the production of SE-33, a truncated retro-analog of LL-37 [91,92]. Initially obtained via chemical synthesis, the peptide lacked a practical application [93]. We subsequently established a bacterial expression system to produce SE-33 in an inactive cassette form, followed by controlled cleavage to its active state. This approach mitigates cytotoxicity toward producer cells and enables large-scale manufacturing. We propose that similar strategies could be adapted for other LL-37 analogs, facilitating their broader integration into clinical practice.

Second, the challenges of in vivo stability, cytotoxicity, and bioavailability must be addressed. Although current strategies such as cyclization, incorporation of D-amino acids, hydrocarbon stapling, and sequence truncation offer considerable improvements, further work is needed to enhance pathogen-specific selectivity. The integration of high-throughput screening technologies and advanced molecular modeling will accelerate the identification of novel cathelicidin analogs with optimized properties.

Third, the development of new nanodelivery systems capable of targeted and stimuli-responsive release (e.g., pH- or temperature-sensitive systems) can ensure local effective concentrations and minimize systemic side effects. At the same time, the complexity and cost of producing certain nanocarriers may limit their large-scale application, and this should be considered during the design process [159].

Beyond nanocarrier-based delivery systems, genetic modification of patient-derived or immunoprivileged cells to express LL-37 and its analogs represents a promising therapeutic strategy. Transplantation of distal airway stem cells engineered to express native LL-37 significantly improved damaged lung repair and protected against bacterial pneumonia and hypoxemia in an animal model [160]. Likewise, human mesenchymal stromal cells (MSCs) modified to express a fusion peptide BPI21/LL-37 exhibited enhanced antibacterial and toxin-neutralizing capacities, reducing organ injury and improving survival in septic mice [161]. Our own studies demonstrated that MSCs engineered to express SE-33 significantly decreased mortality in mice with *S. aureus*-induced pneumonia, reduced bacterial load, and attenuated lung inflammation [162,163]. Taken together, these studies highlight the potential of genetically engineered cells expressing LL-37 or its derivatives as advanced cell-based therapies for severe bacterial infections. This strategy may minimize host cell toxicity, increase targeted delivery efficiency, and potentiate the therapeutic benefits of LL-37 variants.

Finally, the combined use of cathelicidins with conventional antibiotics or immunomodulators represents a promising strategy for overcoming resistance and broadening therapeutic efficacy. However, the pleiotropic nature of the biological functions of the LL-37 complicates its systemic therapeutic use. Therefore, establishing precise therapeutic dosage ranges is critical to minimize the risk of adverse immunological or cytotoxic outcomes [164]. Particular interest should also be given to hybrid peptides, composed of fragments from different natural AMPs, which may provide additive or synergistic effects and increased specificity [121,165,166].

In general, ongoing advances in peptide engineering, combined with the rapid development of computational and synthetic biology tools, are expected to enable the emergence of next-generation cathelicidin-based drugs capable of addressing the global antimicrobial resistance crisis.

## 7. Conclusions

Modified cathelicidins, particularly LL-37 and its derivatives, represent a promising class of potential therapeutic agents that may overcome the limitations associated with both conventional antibiotics and unmodified antimicrobial peptides. The broad-spectrum activity of LL-37, ranging from bacterial membrane disruption and endotoxin neutralization to immunomodulatory and antiviral effects, positions it as a versatile therapeutic agent. Nonetheless, the clinical application of LL-37 is hindered by several critical challenges, including rapid enzymatic degradation, short plasma half-life, low specificity, reduced bioavailability under physiological conditions, and dose-dependent cytotoxic and hemolytic effects towards eukaryotic cells. Additionally, the cost of chemical synthesis of LL-37 and its analogs imposes significant limitations, particularly in regard to the production of full-length peptides.

To address these challenges, a variety of rational design strategies based on structure–activity relationships have been developed. While the present review did not cover in silico modeling approaches of peptide design, it focused on experimentally validated chemical modifications and their functional effects.

Numerous studies, including those highlighted in this review, have demonstrated that site-specific mutagenesis, end-group capping, cyclization, D-amino acid substitutions, and hydrophobic modifications of cathelicidins can lead to the generation of analogs with enhanced antimicrobial efficacy, reduced toxicity, and increased resistance to proteolytic degradation. One of the most promising approaches involves truncation of the LL-37 sequence to form shorter, functionally active fragments (e.g., KR-12, GF-17), which preserve or even exceed the biological activity of the native peptide while exhibiting lower cytotoxicity.

Of particular interest are retro-analogs of LL-37, which maintain the amphipathic α-helical conformation essential for membrane interaction, while showing enhanced resistance to proteolytic degradation and potent antimicrobial activity across a broad spectrum of pathogens. In parallel, the development of nanoscale delivery systems provides an alternative solution to enhance stability, reduce cytotoxicity, improve antimicrobial efficacy against antibiotic-resistant bacteria, and enable targeted delivery to sites of infection or tumor growth.

Further studies into cathelicidin modification, and targeted delivery, coupled with computational modeling and bioinformatics, are crucial to fully understand the therapeutic potential of LL-37 and its analogs and facilitate clinical translation, including optimization of production technologies, evaluation of long-term safety, and the development of effective combination therapies with conventional antibiotics. Addressing these challenges may contribute to advancing cathelicidin-based antimicrobial therapeutics as effective agents in the global effort to combat antibiotic resistance.

## Figures and Tables

**Figure 1 ijms-26-08103-f001:**
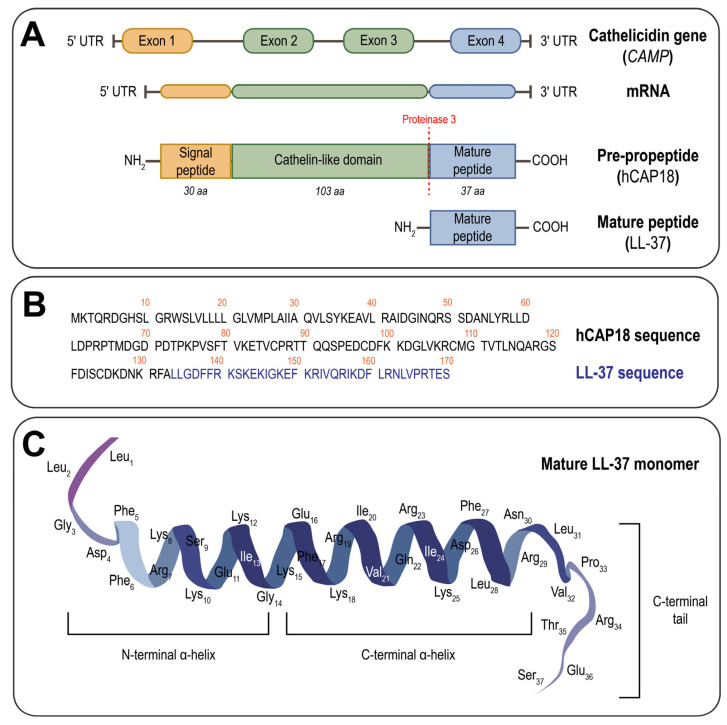
Structural organization of LL-37. (**A**) Schematic representation of the human cathelicidin (*CAMP*) gene comprising four exons. Upon transcription and translation, the gene produces the hCAP18 pre-propeptide, a protein precursor consisting of a signal peptide (orange), a cathelin-like domain (green), and the *C*-terminal LL-37 peptide (blue). Proteolytic cleavage by proteinase 3 (red) releases the active mature LL-37 peptide. (**B**) Amino acid sequence of the full-length hCAP18 pre-propeptide (black), with the LL-37 peptide (residues 134–170, highlighted in blue) obtained by proteolytic cleavage of the *C*-terminal region. (**C**) Three-dimensional structure of LL-37 in its monomeric amphipathic α-helical conformation, including *N*-terminal and *C*-terminal helical domains and a disordered *C*-terminal tail. All amino acid residues are indicated.

**Figure 2 ijms-26-08103-f002:**
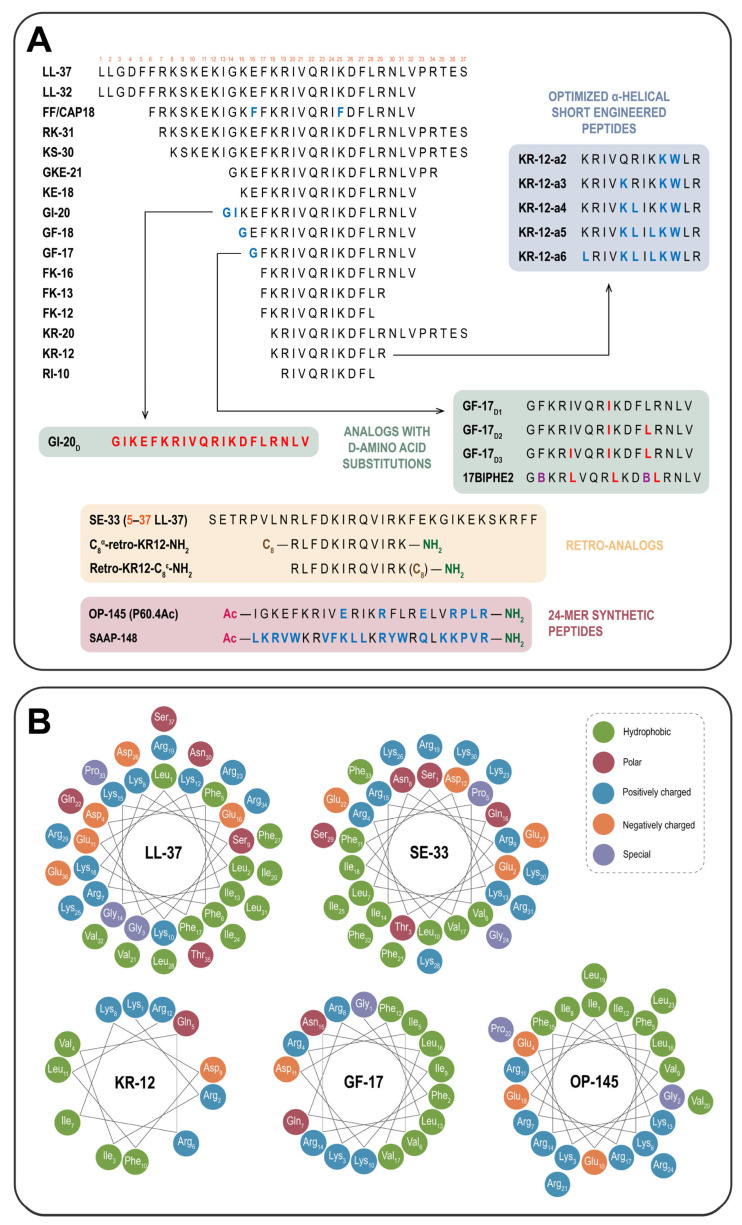
Amino acid sequences and structural characteristics of LL-37 and its analogs. (**A**) Comparative alignment of LL-37 and representative analogs, illustrating sequence modifications such as inversions, substitutions with D-amino acids or biphenylalanine (**B**), acetylation, and truncation. Modifications aim to optimize antimicrobial activity, reduce cytotoxicity, and enhance proteolytic resistance. Color coding: blue—canonical amino acid substitutions; red—substitution with D-amino acids; purple—substitution with biphenylalanine. (**B**) Helical wheel projections (heliograms) depicting the amphipathic nature of LL-37 and its analogs. The spatial distribution of hydrophobic, polar, and charged residues is visualized. Heliograms were created using Protein ORIGAMI web application. Color legend: green—hydrophobic residues; red—polar residues; blue—positively charged residues; orange—negatively charged residues; purple—residues with special properties.

**Figure 3 ijms-26-08103-f003:**
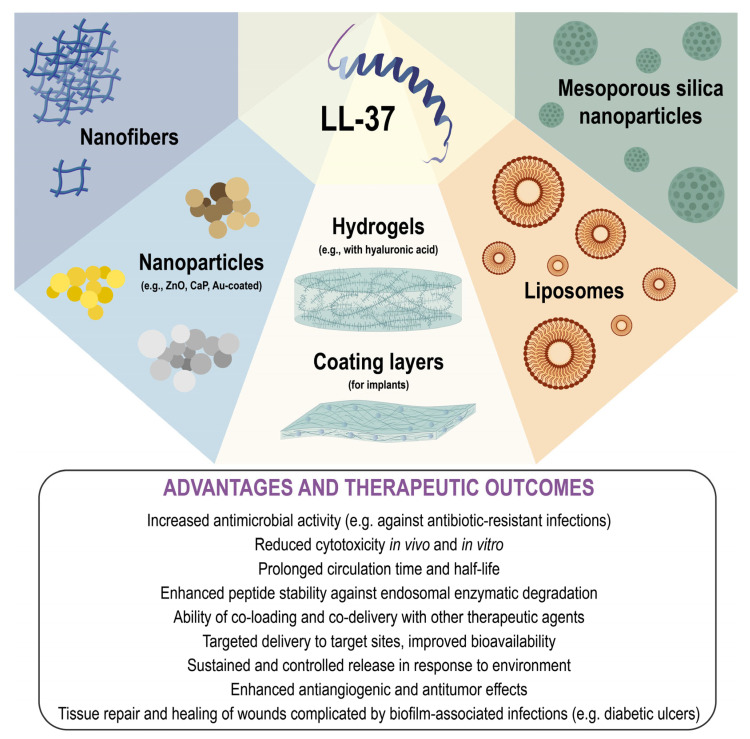
Nanoscale LL-37 delivery systems and their therapeutic potential.

**Table 1 ijms-26-08103-t001:** Functional implications of natural PTMs of LL-37.

Type of PTM	Mechanism	Functional Consequences	References
Citrullination	Enzymatic conversion of Arg to citrulline by PADs Occurs during inflammation and may contribute to the development of autoimmune diseases	Loss of α-helical structure and antibacterial activity Increased antigen presentation Suppression of IFN-I production and B-cell maturation	[54,56,58]
Carbamylation	Non-enzymatic (chemical) reaction with cyanate (MPO-mediated) Lys and Leu convertion to homocitrulline Observed during neutrophilic inflammation	Promotes autoantibody production Maintain innate immune activation (pDCs, B cells) Contributes to SLE pathogenesis	[55,59,60]
Acetylation and formylation	Formation of acetyl and formyl groups at the *N*-terminus of the peptide Observed predominantly in neutrophils	Preserves antimicrobial activity but abrogates autophagy induction Alters immune regulatory functions	[52]

**Table 2 ijms-26-08103-t002:** Strategies for LL-37 modifications and corresponding biological properties.

Modification	Mechanism/Approach	Examples	Biological Properties	References
Truncated analogs	Removal of *N*-terminal hydrophobic residues, shorter bioactive fragments	GF-17, FK-16, FK-13, KR-12, RI-10, etc.	Reduced toxicity Retained activity Improved selectivity and stability	[43,64,65,70,102]
Retro-analog engineering	Sequence reversal while preserving critical residues	SE-33, Retro-KR12-C_8_^ε^-NH_2_, C_8_^α^-retro-KR12-NH_2_	Increased protease resistance Enhanced selectivity	[86,92,93,94]
Point mutations and sequence substitutions	Replacement of hydrophobic residues with hydrophilic amino acids	Positional Q and K mutants of LL-37, derivatives of KR-12 (KR-12-a1 to KR-12-a6)	Reduced toxicity Retained or enhanced antimicrobial activity Decreased hemolytic activity Improved selectivity Increased antitumor activity	[95,97,101]
	Replacement of positively charged amino acids with hydrophobic residues	Glu and Lys to Phe substitutions in FF/CAP18, Ser9 to Ala or Val in LL-37 (LL-23 generation)	Increased antitumor activity Increased antibacterial and immunosuppressive activity	[98,99]
Substitutions of residues with non-canonical amino acid	Targeted incorporation of non-canonical amino acids (e.g., aromatic or β-amino acids)	β-amino acids in LL-37	Increased resistance to proteases Enhanced antimicrobial activity Improved peptide translocation across the bacterial membranes	[108]
Substitutions of L-amino acids with D-enantiomers	Introduction of D-residues at critical positions (e.g., protease-sensitive sites)	GF-17d1-3, GI-20d, 17BIPHE2	Increased resistance to proteases Preservation of activity Decreased cytotoxicity	[51,73,103]
Cyclization	Formation of disulfide bridges or “head-to-tail” cyclization	CD4-PP	Enhanced stability under physiological conditions	[87,88,89]
Hydrocarbon stapling	Introduction of covalent alkyl bridges between side chains of α,α-disubstituted amino acids (e.g., in positions of *i*, *i* + 4 or *i*, *i* + 7)	SLP-51, KR-12(Q5,D9)	Stabilization of α-helix Increased conformational stability Increased protease resistance	[83,129,130]
Terminal modifications	*N*-terminal acetylation, *C*-terminal amidation	OP-145, SAAP-148, AC-1, AC-2	Decreased cytotoxicity Enhanced exopeptidase resistance (prevents aminopeptidase cleavage, reduce carboxypeptidase sensitivity)	[74,75,76,78,79,80,81,84]
Charge and hydrophobicity modulation	Reduction in net positive charge, hydrophobic residue substitution	17F2, positional Q and K mutants of LL-37, KE-18, KR-12	Reduced hemolysis and cytotoxicity with maintained antimicrobial activity Enhanced affinity for bacterial membranes Improved anti-biofilm activity	[67,96,112]
Lipidation	Conjugation with fatty acids (e.g., myristic, lauric, benzoic or octanoic acid)	Laurylated (C_12_-KR12) and myristoylated (C_14_-KR12). KR-12 peptide, Myr-KR-12N, Myr-KR-12C, Retro-KR12-C_8_^ε^-NH_2_	Increased membrane interaction and antimicrobial potency	[86,94,127,128]
Hybridization or dimerization	Fusion of functional domains from multiple peptides or bioactive molecules to combine or enhance effects	LL-37/magainin II fusion; LL-37 conjugation with CpG oligodeoxynucleotides	Increased antimicrobial activity compared to the parental peptides Improve selectivity Enhance activation of innate immunity	[124,125]

## Data Availability

No new data were created or analyzed in this study.

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
