# Peer review of "Antimicrobial Peptides of the Cathelicidin Family: Focus on LL-37 and Its Modifications"

_ijms, 2025, doi:10.3390/ijms26168103_

Round 1

Reviewer 1 Report

Comments and Suggestions for Authors

The manuscript describes antimicrobial peptides of the cathelicidin family. Cathelicidins are peptides with bactericidal properties against a broad spectrum of microorganisms. Only one cathelicidin, hCAP 18, has been demonstrated in humans to date. Under the influence of serine protease 3, hCAP 18 releases an active peptide, LL-37, with bactericidal properties. The authors described peptide optimization strategies: from mutagenesis to computational tools, methods of cathelicidins modification, and various modifications of LL-37. The manuscript was well written and supported by literature data. The text is clear and easy to read.

Specific comments:

  1. It may be worth including references to literature in Tables 1-3.
  2. N- and C- are usually written in italics.
  3. I think Figure 2 is difficult to read.
  4. In Future Prospects, authors should also present their proposal for modifications LL-37 or methods of administration and research.
  5. Complete literary citations according to the journal's guideline, e.g. 9, 143, 157, 161, 162, 170, 178.

Author Response

Comments 1: It may be worth including references to literature in Tables 1-3.

Response 1: We fully agree with this recommendation. Relevant literature references have been added to Tables.

Comments 2: N- and C- are usually written in italics.

Response 2: Thank you for pointing this out. All instances of N- and C- have been italicized throughout the manuscript to conform with standard notation.

Comments 3: I think Figure 2 is difficult to read.

Response 3: We agree with your observation. Figure 2 has been reformatted by adjusting its orientation to improve clarity and readability.

Comments 4: In Future Prospects, authors should also present their proposal for modifications LL-37 or methods of administration and research.

Response 4: We appreciate your suggestion. Additional discussion has been incorporated into Section 6. Future Prospects (Lines 672–685; 697–710).

Comments 5: Complete literary citations according to the journal's guideline, e.g. 9, 143, 157, 161, 162, 170, 178.

Response 5: The reference list has been revised to comply with the journal’s formatting requirements.

Reviewer 2 Report

Comments and Suggestions for Authors

General Comments:

The manuscript provides a comprehensive and well-organized overview of the LL-37 peptide, highlighting its antimicrobial activity, structural modifications, limitations, and future prospects. The topic is both timely and highly relevant, and the authors have made a commendable effort in presenting the current state of knowledge. However, there are several areas where the manuscript would benefit from careful revision and restructuring to improve readability and overall impact. In particular, some sections are repetitive, and certain headings require better alignment with their content. I recommend a minor revision with the specific suggestions outlined below.

Specific Comments:

  1. Repetition and Length Reduction:

The review appears unnecessarily lengthy due to repeated content. To enhance reader engagement, I recommend removing Section 4.2 and incorporating its key points into Table 2. Similarly, the content of Table 1 could be merged into the main text to avoid redundancy and streamline the presentation.

  1. Section and Subsection Titles:

Section 4.3 is currently titled “Basic approach to LL-37,” yet the subsequent subsections (e.g., 4.3.1) focus on modifications aimed at improving LL-37 properties. The title of this section or its subsections should be revised to accurately reflect the content, ensuring better clarity for the reader.

  1. Effect of Modifications on LL-37 Properties:

If the authors intend to elaborate on how modifications influence LL-37’s biological properties, it would be clearer and more impactful to summarize this information in a dedicated table. A suggested title for this table could be: “LL-37 Modifications and Corresponding Biological Properties.”

  1. Re-structuring of Section 4.3.4:

The subsection currently labeled as “4.3.4. Truncated analogs of LL-37” could be elevated to form its own distinct section (Section 4.3) for better emphasis and readability, as it addresses a substantial and independent topic.

Author Response

Comments 1: Repetition and Length Reduction: The review appears unnecessarily lengthy due to repeated content. To enhance reader engagement, I recommend removing Section 4.2 and incorporating its key points into Table 2. Similarly, the content of Table 1 could be merged into the main text to avoid redundancy and streamline the presentation.

Response 1: We agree with this suggestion. Due to repeated content, Sections 2 and 3 were removed, with relevant information redistributed into other appropriate sections of the manuscript. Table 1 was also removed to eliminate redundancy, as the associated content was restructured and integrated into the main text.

Comments 2: Section and Subsection Titles: Section 4.3 is currently titled “Basic approach to LL-37,” yet the subsequent subsections (e.g., 4.3.1) focus on modifications aimed at improving LL-37 properties. The title of this section or its subsections should be revised to accurately reflect the content, ensuring better clarity for the reader.

Response 2: Thank you for this observation. The main title of Section 4 and its subsections have been revised to better reflect their content and to improve clarity for the reader.

Comments 3: Effect of Modifications on LL-37 Properties: If the authors intend to elaborate on how modifications influence LL-37’s biological properties, it would be clearer and more impactful to summarize this information in a dedicated table. A suggested title for this table could be: “LL-37 Modifications and Corresponding Biological Properties.”.

Response 3: We appreciate this valuable recommendation. A new table summarizing LL-37 modifications and their corresponding biological properties has been added at the end of Section 4 (Line 592).

Comments 4: Re-structuring of Section 4.3.4: The subsection currently labeled as “4.3.4. Truncated analogs of LL-37” could be elevated to form its own distinct section (Section 4.3) for better emphasis and readability, as it addresses a substantial and independent topic.

Response 4: This recommendation has been fully implemented. Section 4 has been substantially restructured so that Sections 4.1 and 4.2 are now dedicated to “Shortened derivatives: truncated LL-37 analogs” and “Sequence-reversed variants: retro-analogs of LL-37”, respectively, ensuring better emphasis and readability.

Reviewer 3 Report

Comments and Suggestions for Authors

IJMS 3794598

Title: Antimicrobial Peptides of the Cathelicidin Family: Focus on 2 LL-37 and Its Modifications

Here are my suggestions

Major:

-The review is too long. It contains a lot of information and is difficult to follow. I suggest deleting point 3, Approaches to the modification of cathelicidins, and focusing on the strategies for LL-37 modification (Point 4). Many examples of point 3 are repeated in point 4.

-Are any in vivo studies for LL-37 or its variants? Please add this information.

Author Response

Comments 1: The review is too long. It contains a lot of information and is difficult to follow. I suggest deleting point 3, Approaches to the modification of cathelicidins, and focusing on the strategies for LL-37 modification (Point 4). Many examples of point 3 are repeated in point 4.

Response 1: We fully agree with this suggestion. In addition to removing Section 3, we also deleted Section 2. Relevant content from these sections was selectively restructured and integrated into appropriate parts of the manuscript. Repetitive examples were reviewed or eliminated to improve focus and readability.

Comments 2: Are any in vivo studies for LL-37 or its variants? Please add this information.

Response 2: Thank you for highlighting this important point. We have revised the existing discussions in the manuscript to clearly indicate which studies involved in vivo experiments. Furthermore, we have added new references and relevant information on in vivo studies of LL-37 and its derivatives to provide a more comprehensive overview (Lines 106–116; 329–331; 567–569; 583–586; 661–663; 672–677; 697–710).

Reviewer 4 Report

Comments and Suggestions for Authors

The paper is suggested for publication after major revisions.

Author Response

Comments 1: The paper mainly focused on LL-37 and its modification, the manuscript should detailly describe the key structure factor of LL-37 for its antimicrobial activity, such as positive charges, hydrophobicity, secondary structure, amphiphilicity and so on. However, the relevant contents are simple and rough in 4.1 section. In addition, there are some redundant contents in this manuscript, for example, “2. Strategies for peptide optimization: from mutagenesis to computational tools”, which should be deleted in the manuscript.

Response 1: We fully agree with your suggestions. Section 2 was removed, and relevant information was redistributed across later sections. We also created a dedicated section (Section 2. Structural and functional features of LL-37; Lines 129–204), which has been expanded to include a detailed description of key structural properties underlying biological activity of LL-37.

Comments 2: “3.2. Strategies for reducing toxicity and enhancing stability” should be divided into two titles to clearly describe each aspect, but they were mixed together in the manuscript.

Response 2: Thank you for this recommendation. The section has been completely restructured to focus more specifically on LL-37 modifications. Relevant content has been reorganized and incorporated into the new Sections 4.3.1 and 4.3.2, with additional discussion and supporting references.

Comments 3: In “4.3 Basic approaches to LL-37 modification”, the contents would be clear and vivid if the paper is expressed according to different strategies, such as modulation of physicochemical properties (such as hydrophobicity and charge), modifications of amino acid sequence (e.g., enantiomer substitution, cyclization, terminal modification), design of truncated forms or retro-analogs, as well as nanostructured delivery design.

Response 3: We have adopted your suggested structure. Section 4 is now organized according to modification strategies rather than by final biological effects, which has improved the clarity and logical flow of the manuscript.

Comments 4: In “4.3.2 Reduction of cytotoxic and hemolytic activity”, KR-12 series derivatives did not suggest that “lowering hydrophobicity leads to reduced cytotoxicity……towards mammalian cells [109]”, and more reports should be offered to explain the effect of net charge and hydrophobicity on peptide activity. The similar question was also found in “4.3.3 Enhancement of antimicrobial and other biological efficacy”, for example, “Hydrophobicity and amphipathicity are critical for the membranedisruptive action of LL-37…..”, which should be further extended and argued in the manuscript.

Response 4: We have expanded the relevant discussions and added additional references to better support and critically evaluate these points. These revisions can be found in Section 4.3.2 (Lines 460–482; 496–511).

Comments 5: In Page 15, what is the structure of the variant SLP-15, LL37-1, ACL-37-1/-2 or DL-37-2(D)? What are their structural differences compared to parent peptide? This information was important to understand the structure-activity relationship.

Response 5: The requested structural details and comparisons have been added (Lines 357–362).

Comments 6: The paper highlights the nanoscale delivery systems. In this section, an additional figure should be added to better summarize “4.3.6 Nanoscale delivery systems for LL- 37”. (7) The relevant references should be added in Table 1-3.

Response 6: We appreciate your suggestion. We have added Figure 3 to Section 5. Nanoscale delivery platforms for LL-37 and its derivatives to provide a concise visual summary.

Comments 7: The relevant references should be added in Table 1-3.

Response 7: We agree with this recommendation. References have been added into Tables as suggested.

Round 2

Reviewer 3 Report

Comments and Suggestions for Authors

The manuscript demonstrates an exhaustive search of the bibliography and reflects the authors´ knowledge about human cathelicidin LL-37. The authors also included their results and experience with the peptide. The manuscript has been improved in this version. Now it focuses on human cathelicidins, making it easier to understand.   The information about in vivo assays was added as requested. 

Reviewer 4 Report

Comments and Suggestions for Authors The authors have adequately addressed my concerns in the revised manuscript.